# OpenReview forum: "Breaking the Chain: A Causal Analysis of LLM Faithfulness to Intermediate Structures"
_ICLR.cc/2026/Conference — Submitted to ICLR 2026_

### Official Review · Reviewer_ewGT · 2025-10-30

**Soundness:** 4
**Presentation:** 3
**Contribution:** 4
**Rating:** 8
**Confidence:** 4

**Summary:**

This paper studies whether LLM's answer faithfully follows from its reasoning by perturbing the reasoning and seeing if the answer is then still consistent with the new reasoning. While previous studies have learned that LLM reasonings are not faithful to its answer, they have not studied this with a causal analysis. This paper uses four datasets that have structured intermediate reasonings such as rubrics, and change the rubric scores and see if the answer follows from the new score or the previous. They did these different interventions: HSVT where they hold the intermediate structure M and change the input, such that relying on the structure should give the same answer despite the change in input; LEC, where individual elements of M are fliped, expecting the answer to change accordingly, and GEC, where all valid support should be flipped. And they experimented with both the model's original reasoning and the ground truth one. Although on three datasets the ground truth one gives higher accuracy than the model generated one, but surprisingly it's the opposite for Entailment Bank. This is likely because using the gold intermediate structure was out of distribution and thus made the performance worse. They did find a correlation between structure prediction accuracy and final answer accuracy, and GEC is more consistent that LEC. The findings from the paper suggest that if we were to do AI and human collaboration by having humans intervening the intermediate steps, LLMs are currently still unreliable because they often don't follow from the new reasoning steps.

**Strengths:**

1. The paper is well-written and easy-to-follow.
2. The question being studied is important: to causally analyze whether the model can follow any reasoning. If not, then we can't even expect it to have improved performance when intervened upon.
3. The experiments are comprehensive. The adaptations of the datasets for the use in this paper is great.

**Weaknesses:**

1. No error bars in the bar charts.

**Questions:**

1. For Figure 6, it seems that HSVT is quite high for most models. Could we say that, in general, if the intermediate structure is in-distribution, then the answer might be more consistent with the structure? In HSVT, the intermediate structure didn’t change, and changing the input X is farther away and thus maybe has less impact on the final prediction. Meanwhile, in the other cases (global/local edits), the intermediate structure is changed and thus more out-of-distribution, and it’s closer to the answer, which induces more uncertainties. What is your intuition on this?
2. For EntailmentBank, it seems that if a model is more faithful on gold, then it is less faithful on predicted intermediate structure, and if it's more faithful on predicted, then less faithful on gold (Llama-3.2-1B is reversed from the other models but the trend is the same). Do you have any intuitions on this?

---

> ### Author Response · Authors · 2025-11-23
>
> Thank you for your thoughtful and positive review. We greatly appreciate your careful reading of our paper and we are grateful that you have highlighted the clarity of our presentation, the significance of the question we address, and the comprehensiveness of our experimental design, including the adaptation of the datasets.
> Regarding the missing error bars, we will include confidence intervals for all relevant comparisons in the revised version, placing detailed values in the appendix to keep the figures readable.
> As for your questions, thank you for your interest in these behaviors. We address each point below.
> 1. Thank you for raising this point. The distributional explanation you suggest is indeed plausible, and we agree that out-of-distribution edits to M could in principle induce instability. However, our interpretation is that the observed high HSVT and low LEC/GEC are better explained by a deeper causal asymmetry in how models use input X and mediator M. When X is changed but M remains fixed, the model usually retains its original decision, rather than recalculating it from the input, which leads to high HSVT. However, this reflects the stability of the original prediction, rather than mediated reasoning. When M is changed, the model often keeps the same decision even though the edited structure should lead to a different conclusion, which is what drives the low LEC and GEC values. In both situations, the underlying behavior of the model is the same. It anchors on its initial prediction and treats both X and M as context, rather than causal drivers of Y. Therefore, edits to M are not considered meaningful counterfactual interventions within the model's reasoning. This pattern does not indicate that unedited M is "in-distribution" while edited M is "out-of-distribution". Instead, it shows that the model does not use M as a mediator, either when X changes or when M changes. Instead, it reflects the model's tendency to rely on its internal shortcuts for producing Y. Low LEC and GEC indicate that even significant edits to M cannot override this reliance on internal knowledge.
>
>
> 2. It is a very good question. We would first note that the faithfulness scores on EntailmentBank are quite low across most models (see Figure 6), so we are cautious about inferring stable trends from this dataset alone. That said, our intuition is consistent with the explanation in point 1. When the model generates its own proof, that structure often reflects the latent shortcuts it actually relies on, so interventions on predicted proofs disrupt these shortcuts more directly, leading to higher faithfulness. In contrast, gold proofs do not contain these internal shortcuts, and the model may therefore treat them as external context, which makes edits less impactful. However, given the overall complexity of EntailmentBank’s proof structures, the opposite pattern can also occur: gold proofs are semantically correct and structurally well-formed, whereas our experiments show that predicted proofs may contain inconsistencies or missing steps (especially for smaller models). In such cases, interventions on gold proofs modify the reasoning in a cleaner and more interpretable way, while edits to predicted proofs may not meaningfully change the model’s internal decision. This can make faithfulness appear higher on gold proofs.

---

> ### Author Response · Authors · 2025-11-27
>
> Dear Reviewer, we have updated our PDF with your concerns and questions, please let us know if you still have any concerns or questions, and whether our comments have affected your overall assessment of our paper.

---

### Official Review · Reviewer_zgVw · 2025-10-30

**Soundness:** 2
**Presentation:** 3
**Contribution:** 3
**Rating:** 2
**Confidence:** 3

**Summary:**

This paper explores the role of structured CoT in LRM response generation. The authors find that while models rely on the intermediate structure generated in CoT, changes to such structure does not always lead the model to produce the logically sound conclusion from the edited structure. Additionally, models respond even less to changes in ground truth intermediate structure compared to ones they generate, even if their structure is wrong.

**Strengths:**

The paper is well-written and clear, and addresses an interesting problem of the logical consistency of LRMs and their faithfulness to their own CoT explanations. The focus on structured CoT makes evaluations more clear since the ground truth answer is usually clear.

**Weaknesses:**

My main issue is the following:

If the generated structure (e.g., rubric) is edited but the prompt (e.g., the student's answer) is not edited, doesn't this generate a contradiction between the prompt and the CoT? If this is the case, could the authors clarify what the expected output of the model should be? If the edits are introducing a logical contradiction then, to me, it isn't clear that the model should be expected to answer according to the content of the rubric (this would require it to actively ignore the content of the prompt). I feel that the significance of the results in this work rests heavily on  whether the observed behavior is surprising/unexpected, and this is the main reason for my low score.

Other thoughts:
- In the HSVT setting (Fig 5), it is not clear which (if any) of the prompt-level interventions lead to a logical contradiction with the structure. e.g., simple paraphrasing probably does not create a contradiction, but global edits (replacing the original student's answer with a different student's answer) may or may not contradict the structure (maybe the replaced answer would generate the same rubric/score). This seems like the crux of the analysis in my opinion and does not appear to be clearly evaluated by the work currently.
- One of the main outcomes of the work is that when the structure is changed, the model often does not change its final answer to be congruent with these changes. When the intermediate structure is changed but the prompt is not, and the structure would imply a different final answer than the prompt (e.g., the edited rubric implies the score for this question is 4 but a correct rubric for the answer in the prompt would score the question 2), does the model answer 2 or 4? Or neither? i.e., is the model completely wrong or is it just ignoring a rubric that is logically inconsistent with the answer?
- More general question (won't affect my score, just curious what the authors think): Is there a way to formalize the distinction between the types of prompt edits that should cause a change in the model's downstream answer vs edits that a good model would be robust to? e.g., we know that paraphrasing or rewriting keeps the same information content/logical conclusion, while the global edits of replacing the whole answer likely changes the information content substantially.

Overall, I think the premise of this work is interesting and I would be amenable to raising my score if my concerns are addressed, especially the key concern at the top.

**Questions:**

Most of my major questions are in 'weaknesses'.

Minor questions:
- Could the authors provide more detail about the different benchmark datasets? Only 2 example prompts were given.
- Another conclusion from the paper is about the models' faithfulness to gold vs predicted structures. This is an interesting result, but isn't there usually a single way to populate the structure correctly (e.g., thinking about the rubric example)? If so, isn't this more about the predicted structure being wrong? How often is the predicted structure wrong?

---

> ### Author Response · Authors · 2025-11-23
>
> Hi, thank you for taking the time to review our paper.
>
> ### Key Concern
>
> Yes — in different settings we intentionally introduce contradictions between the input (e.g., student answer or claim text) and the edited intermediate structure. This is done by design. Our goal is to measure whether the model treats the structure as a **causal mediator** rather than just contextual text.
>
> Under this causal framing, when the structure is changed, we expect the model to update the final answer accordingly, even if the input remains unchanged. This reflects real human–AI collaboration scenarios (e.g., a teacher corrects part of a rubric expecting the score to change without rewriting the student answer, or a programmer corrects the model plan before generating the code). Therefore, the expected output is the one implied by the edited structure, not the original input.
>
> The surprising result is that models often ignore these edits even when explicitly instructed to base the answer on the intermediate structure. This behavior suggests that mediator structure is not treated as a true decision pathway.
>
> ### Why This Behavior Is Unexpected
>
> We believe it is unexpected because:
> 1. The model is prompted to reason through the structure and base decisions on it.
> 2. The structure is visible and explicit.
> 3. The structure encodes the full logic needed for the final output.
>
> We literally instruct the model in RiceChem for example:
>
> ```
> * You must fill the checklist (True/False) strictly based on the student's answer.
> * The final grade equals the sum of the weights of the items marked True.
> ```
> Thus, a faithful model should follow the mediator even when it is wrong. Our experiments show the opposite — the model preserves the original output or fails to process the mediator correctly.
>
> ### Soft vs. Hard Interventions
>
> Regarding your point that paraphrasing may not create contradictions but global edits do:
>
> - **RiceChem and TabFact** — we apply **hard interventions** by replacing the student answer or associated table context (which may introduce contradictions).
> - **Averitec and EntailmentBank** — we apply **soft interventions** such as paraphrasing or entity renaming, which preserve meaning.
>
> As we show in **Figure 5**, hard interventions result in lower HSVT because the model does ground on X. We added this explicitly in the last paragraph of Section 5.2.
>
> ### When Structure Implies a Different Outcome
>
> Example:
> If the edited structure implies score **4** but the original input implies score **2**, we treat **4** as the expected answer. If the model outputs the original answer, it is not being "robust" to intervention — it is ignoring the structured mediator and relying on the original input or latent knowledge. It is indeed interesting when the model answers incorrectly, what does it predict? We measure what the model predicts when the mediator is modified in LEC and GEC settings.
>
> | Type / Structure                          | Description | LEC | GEC |
> |-------------------------------------------|-------------|-----|-----|
> | Back2Previus Acc (Predicted Structure)       | How often the model predicts the same score as before intervention (Accuracy) | 0.273 | 0.058 |
> | Back2Previus Acc (Gold Structure)            |  | 0.340 | 0.131 |
> | Expected2Predicted MAE (Predicted Structure) | Error relative to correct score after intervention (MAE) | 0.962 | 1.589 |
> | Expected2Predicted MAE (Gold Structure)      |  | 1.275 | 1.661 |
>
> #### Interpretation
>
> **Back2Previus Acc**
> - In LEC, the model reverts much more to the previous score than in GEC.
> - Predicted structure is more susceptible to intervention changes (changes score less frequently back to the previous score).
> - This supports our claim that the model is more faithful to *generated* mediators.
>
> **Expected2Predicted MAE**
> - Error under predicted structure interventions is lower than under gold structure interventions.
>
> ### Gold vs. Predicted Structures
> Regarding gold vs. predicted structures: our claim is not only that predicted structures may contain errors, but that models are often more responsive to edits in their self-generated structures than in gold ones. We observe this by measuring how often interventions alter outputs in both settings (Section 5.2; Appendix 2). Appendix A.3 reports structure and score accuracy, showing that even correct structures do not reliably yield correct scores—indicating that models do not fully use the structure as a causal mediator.
> As shown there, correct structure does not always imply correct score, indicating that the model does not fully utilize structure as mediator, supporting our claims.
>
> We have also included full prompt templates in the appendix to make the experimental setup clearer.
>
> Thank you again for the helpful feedback. We believe these clarifications make the contribution more explicit, especially regarding how contradictions are used to test causal mediation rather than model correctness.

---

> ### Author Response · Authors · 2025-11-27
>
> Dear Reviewer, we have updated our PDF with your concerns and questions, please let us know if you still have any concerns or questions, and whether our comments have affected your overall assessment of our paper.

---

### Official Review · Reviewer_dryL · 2025-10-31

**Soundness:** 3
**Presentation:** 3
**Contribution:** 2
**Rating:** 6
**Confidence:** 3

**Summary:**

The paper investigates the causal role of intermediate structures like rubrics and checklists that LLM generate as part of their decision-making process. The paper propose an evaluation protocol designed to test whether these structures act as true causal mediators for the final decision or decorative by-products. The protocol introduces three metrics which are HSVT, LEC, and GEC. Applying this protocol across various models and benchmarks, the paper reports a systematic gap that the model appear consistent with their structures when the input is pertubed but frequently fail to update their output when the structure itself is directly edited.

**Strengths:**

1. The paper is well-written and easy to follow.
2. The finding were clearly stated and demonstrated, with broader implications for understanding reasoning models.
3. The evaluation is comprehensive, testing 9 LLMs across 4 diverse benchmarks.

**Weaknesses:**

1. The experiment result should show that it is statistically significant (e.g., use CI, paired t-test) not the point-estimate averages in Figure 3 and 4.
2. The metric appear to be binary, which could be a limitation for quantitative task.
3. Faithfulness metrics partly conflate invariance with mediation, which I think the author should specify the definition of 'faithfulness'.
4. The paper should evaluation additional prompting beyond few-shot, as improved edit sensitivity under stronger prompt would suggest that the reported causal mediation gaps may stem from prompt under-specification.
5. The paper need to validate the core assumption that the interventions are not OOD.
6. Even though the paper compared with various LLMs, it omit leading proprietary models like GPT or Claude limiting the generality.

**Questions:**

Look at the weaknesses

---

> ### Author Response · Authors · 2025-11-23
>
> Hi! Thank you for your review. We appreciate your positive assessment of the clarity of our paper, as well as the recognition of the breadth of our evaluation across diverse LLMs and benchmarks.
>
> Regarding the weaknesses:
>
> 1. We appreciate the suggestion. While Figure 3 and Figure 4 report aggregate trends, Table 1 already provides confidence intervals for the key correlation results. The full set of CI values for gold vs. predicted performance does not fit cleanly into the figure layout, but we will include them in the appendix and add a brief note in the main text to make this explicit.
> For Figure 3, we additionally run Wilcoxon signed rank test to check whether gold structure improves performance compared to predicted structure across models and dataset. It shows that gold structure delivers statistically significant improvement in performance (p=0.002, with sample size 36 = 4 datasets x 9 models). For Figure 4, we additionally measure standard error of performance estimate for each model and structure prediction type. We find out all individual differences are statistically significant under two proportion z-test:
>
> Llama 3.2 1B:
> - Predicted: performance = 0.812, SE = 0.0212
> - Gold: performance = 0.694, SE = 0.0250
> - Difference: -0.148, SE(diff) = 0.0255
> - Z-statistic: 3.567, p-value: 0.0004
> - Statistically significant difference (Predicted > Gold, p < 0.05)
>
> Falcon 3 7B:
> - Predicted: p = 0.988, SE = 0.0059
> - Gold: p = 0.926, SE = 0.0142
> - Difference: 0.062, SE(diff) = 0.0156
> - Z-statistic: 3.985, p-value: 0.0001
> - Statistically significant difference (Predicted > Gold, p < 0.05)
>
> Falcon 3 3B:
> - Predicted: p = 0.976, SE = 0.0083
> - Gold: p = 0.653, SE = 0.0258
> - Difference: 0.323, SE(diff) = 0.0298
> - Z-statistic: 10.835, p-value: 0.0000
> - Statistically significant difference (Predicted > Gold, p < 0.05)
>
>
> 2. In our framework, faithfulness is defined as strict causal mediation: once the mediator M is present, the model should base its final decision solely on M. Under this definition, any residual dependence on X – including cases where the model “softens” or partially overrides the effect of an edit to M – already constitutes a violation of faithfulness. For this reason, the notion of a “partially faithful” response is not well-defined in our setting: faithfulness is about whether the causal influence of M is respected, not about the magnitude of numerical changes in the downstream task. Even when the downstream output is quantitative (e.g., a score in RiceChem), the correctness criterion for faithfulness is categorical – does the model follow the mediator or not? – and therefore binary metrics naturally match our causal definition.
>
> 3. Our notion of faithfulness is explicitly causal mediation: the model’s final decision should be determined by the intermediate structure M once M is present. In this setup, a faithful model should no longer rely on the original input X after generating or receiving M, as M is intended to contain all task-relevant reasoning that guides the final prediction. HSVT (invariance) is therefore not a competing notion but a special case of testing mediation: if M is the mediator, then keeping M fixed while varying X should not change the outcome. Thus the metrics do not conflate two concepts — they quantify complementary facets of the same underlying causal requirement that the model’s prediction flows through M. Put more simply, HSVT asks “should the model not react to this change?”, whereas global edits ask “should the model react to this change?”. While the prompts may look superficially similar, the causal semantics they probe are fundamentally different.
>
> 4. Thank you for this suggestion. We agree that stronger prompting may shed additional light on whether the observed mediation gap is due to prompt under-specification. In the revised version, we run an additional set of experiments where the prompt explicitly warns the model that interventions on the mediator may occur and provides concrete examples of such edits. This version of the prompt also clearly instructs the model to treat the mediator as the sole source of task-relevant reasoning and to update its final decision accordingly when edits are introduced.
>
> For each faithfulness metric (HSVT, LEC, GEC) and setting (gold/predicted structure) we run Wilcoxon signed rank test between default and detailed prompt. We find that detailed prompt results in statistically significant regression in faithfulness in HSVT, gold setting ($p=0.0013$) and statistically significant improvement in faithfulness in GEC, gold ($p=0.039$) and GEC, predicted ($p=0.027$) settings. Three other configurations (HSVT predicted, LEC gold, LEC predicted) do not show statistically significant difference. Test sample size is 36 (4 datasets x 9 models).

---

> ### Author Response · Authors · 2025-11-23
>
> 5. Our interventions are schema-preserving and therefore do not introduce structural out-of-distribution inputs. Our setting involves no task-specific fine-tuning, and the prompts explicitly define the expected mediator schemas and clarify that imperfect or partially malformed structures may appear. The interventions stay within these schemas: rubric flips keep the rubric template intact, proof edits preserve the tree structure, and DSL edits preserve the grammar. The model is not exposed to syntactically or structurally unfamiliar inputs.
> At the semantic level, the intervening structures are also not foreign to the model. In the predicted setting, models naturally generate imperfect, contradictory, or partially broken mediators of the same kinds as those we edit. This shows that such variants lie within the model’s operational distribution. Our edits do not introduce qualitatively new artifacts but modify existing structures in a controlled way to change their logical implications.
>
> 6. We appreciate the reviewer’s point about proprietary models. While our primary focus was on open models to ensure reproducibility and consistent access, we agree that including a large state-of-the-art system would strengthen the generality of the findings. Despite practical limitations on proprietary model inference, we will apply our protocol to the gpt-oss-120b model and report the corresponding results in the revised version.

---

> ### Author Response · Authors · 2025-11-27
>
> Dear Reviewer, we have updated our PDF with your concerns and questions, please let us know if you still have any concerns or questions, and whether our comments have affected your overall assessment of our paper.

---

> ### Author Response · Authors · 2025-11-28
>
> Update on point 6:
>
> We conducted our experiments on the Qwen3-235B-A22B model. The results are in Tables:
>
> **Performance:**
> | Dataset   | With gold structure                      | With predicted structure                                   |
> |-----------|----------------------------------------------|-----------------------------------------------------------------|
> | TabFact   | label_match = 0.706 ± 0.456                  | expression_match = 0.394 ± 0.489; label_match = 0.671 ± 0.470 |
> | AVeriTeC  | verdict_match = 0.624 ± 0.484                | structure_match = 0.848 ± 0.359; verdict_match = 0.944 ± 0.230|
> | RiceChem  | score_match = 0.486 ± 0.500                  | checklist_match = 0.342 ± 0.474; score_match = 0.285 ± 0.451  |
>
> **Faithfulness**
> | Dataset  | HSVT (gold)   | HSVT (pred)   | LEC (gold)    | LEC (pred)    | Global (gold) | Global (pred) |
> | -------- | ------------- | ------------- | ------------- | ------------- | ------------- | ------------- |
> | TabFact  | 0.571 ± 0.495 | 0.546 ± 0.498 | 0.883 ± 0.321 | 0.879 ± 0.326 | 0.908 ± 0.289 | 0.914 ± 0.280 |
> | AVeriTeC | 0.517 ± 0.500 | 0.657 ± 0.475 | 0.096 ± 0.294 | 0.238 ± 0.426 | 0.750 ± 0.433 | 0.929 ± 0.258 |
> | RiceChem | 0.772 ± 0.419 | 0.737 ± 0.440 | 0.372 ± 0.483 | 0.329 ± 0.470 | 0.624 ± 0.484 | 0.726 ± 0.446 |
>
>
> The overall trends are consistent with our previous findings, but there are some notable differences. First, the HSVT scores were lower for the larger model, indicating that it relies more strongly on direct information from the input X and is less influenced by the mediator M. This aligns with our hypothesis that larger models may have more direct predictive pathways that bypass the mediator, and that these pathways become more prominent with scale. Second, on RiceChem and AVeriTeC, the LEC scores for the 235B model were smaller than for the 2–8B models, suggesting that larger models are more likely to ignore localized but meaningful changes in the mediator. This also supports the idea of shortcut-heavy behavior and the greater stability expected from larger models. In contrast, GEC was significantly higher for the large model, indicating that it not only orientates itself around the mediator M, but also responds more systematically to major changes in the mediator compared to smaller models.. Finally, the performance and accuracy scores on TabFact are significantly higher for larger models than for smaller ones, which supports the hypothesis that larger models are better at treating verifier queries as executable programs rather than just random context.
>
> These findings suggest that scaling up reinforces the use of shortcuts and increases local robustness, while also making the impact of significant changes in the mediator more noticeable than in smaller models. We plan to continue this analysis on other large-scale architectures, such as LLaMA and Falcon, to determine whether these trends hold true across different model families.

---

> ### Author Response · Authors · 2025-12-03
>
> We also conducted experiments for Llama-3.1-70B-Instruct. the results are in tables:
>
> **Performance:**
> | Dataset        | Gold structure                 | Predicted structure                                                                           |
> | -------------- | ------------------------------ | --------------------------------------------------------------------------------------------- |
> | TabFact        | label_match = 0.613 ± 0.487    | expression_match = 0.289 ± 0.453; label_match = 0.771 ± 0.420                                 |
> | AVeriTeC       | verdict_match = 0.972 ± 0.165  | structure_match = 0.944 ± 0.230; verdict_match = 0.966 ± 0.180                                |
> | RiceChem       | score_match = 0.533 ± 0.499    | checklist_match = 0.361 ± 0.480; score_match = 0.344 ± 0.475                                  |
> | EntailmentBank | score_match = 0.997 ± 0.054    | proof_match = 0.003 ± 0.054; sentence_set_match = 0.088 ± 0.284; score_match = 0.460 ± 0.498  |
>
> **Faithfulness:**
> | Dataset        | HSVT (gold)    | HSVT (pred)    | LEC (gold)      | LEC (pred)      | Global (gold)     | Global (pred)   |
> | -------------- | -------------- | -------------- | --------------- | --------------- | ----------------- | --------------- |
> | TabFact        | 0.411 ± 0.492  | 0.289 ± 0.453  | 0.628 ± 0.483   | 0.617 ± 0.486   | 0.726 ± 0.446     | 0.739 ± 0.439   |
> | AVeriTeC       | 0.983 ± 0.129  | 0.983 ± 0.129  | 0.402 ± 0.490   | 0.398 ± 0.489   | 1.000 ± 0.000     | 1.000 ± 0.000   |
> | RiceChem       | 0.844 ± 0.363  | 0.812 ± 0.391  | 0.368 ± 0.482   | 0.337 ± 0.473   | 0.487 ± 0.500     | 0.570 ± 0.495   |
> | EntailmentBank | 0.994 ± 0.077  | 0.981 ± 0.137  | 0.0590 ± 0.236  | 0.0598 ± 0.237  | 0.00590 ± 0.0766  | 0.0256 ± 0.158  |
>
> LLaMA-3.1-70B-Instruct paints a different picture. Its faithfulness profile is much closer to that of 2–8B models: HSVT scores are generally high, especially on AVeriTeC and EntailmentBank, indicating a strong orientation toward the mediator, while LEC and GEC remain low or moderate and, in the case of EntailmentBank, GEC is very close to zero. Thus, LLaMA-3.1-70B-Instruct behaves more like a scaled-up version of the smaller models in our study, whereas Qwen3-235B-A22B-Instruct exhibits a more pronounced shortcut-like pattern.
>
> Taken together, these results suggest that scaling can amplify different aspects of a model’s inductive bias: in some architectures (e.g., Qwen) it appears to strengthen direct pathways from X to the prediction, while in others (e.g., LLaMA) faithfulness remains in the same qualitative regime as for smaller models.

---

> > ### Author Response · Authors · 2025-12-03
> >
> > We have updated our PDF with new results.

---

### Official Review · Reviewer_NWDA · 2025-11-06

**Soundness:** 1
**Presentation:** 1
**Contribution:** 2
**Rating:** 2
**Confidence:** 3

**Summary:**

This paper proposed an evaluation method to assess whether LLMs faithfully relied on intermediate reasoning structures (e.g., rubrics, checklists, proof graphs) during inference. Specifically, the authors observed whether model outputs changed by modifying either the intermediate structures or the input text, thereby demonstrating whether model decisions are driven by these structures. Experiments showed that while LLMs relied on intermediate structures, this dependency exhibited weak causality. Furthermore, models appeared to be more faithful to their self-generated structures than to gold-standard ones.

**Strengths:**

This paper aimed to explore the faithfulness of LLM-generated content to intermediate reasoning structures, which held practical significance for understanding LLM causal reasoning capabilities.

**Weaknesses:**

$\bullet$  The main limitation lies in the lack of a clear research objective and insufficient consideration of the complexity of natural language.

(1) For autoregressive language models, what exactly does the intermediate reasoning representation $M^* \neq M$ refer to? What level of discrepancy is being considered, token-level or semantic-level? If token-level, $M^* \neq M$ does not necessarily imply the final decision $Y^* \neq Y$, so Eq. (2) may not hold. If semantic-level, how is semantic inconsistency defined and quantified? Semantic variation can range from minor paraphrases to contradictory meanings. Even when $M \neq M'$, both $Y=Y'$ and $Y\neq Y'$ (as in adversarial examples) are possible. Which case do the authors intend to capture?

(2) How is unfaithfulness defined? In most natural-language tasks, “unfaithful reasoning” is not well-defined. Even in mathematical or programming tasks, multiple reasoning paths (M) may yield the same outcome. Why should obtaining the same Y from the same X but a different M necessarily imply unfaithfulness? The transition from $p_\theta (Y∣X,M)$ to $p_\theta (Y^* ∣X, M^* )$  with $M^* \neq M$ does not strictly guarantee that   $Y \neq Y^* $.

(3) Measuring causality through output change is valid only under restricted settings. For many tasks (e.g., open-domain dialogue), what constitutes causality is unclear. Even in tasks with well-defined causal structures, modifying M may not necessarily change the output Y.


$\bullet$ The paper should provide explicit mathematical definitions and quantitative measures for key concepts. What is LLM unfaithfulness, and how is it measured? What qualifies as a *logically significant edit*, and how are modifications to M quantified? How is *causality* defined across tasks, and how can it be measured?

$\bullet$ The authors argue that “If an LLM is faithful to its intermediate structures… then logically significant edits should change its final decision.” This statement is an assumption, not a fact. Modifications to intermediate structures may alter the probability distribution of the model's output, but they do not necessarily change the final decision. For instance, the top-probability token might remain the same even if its confidence decreases sharply. Should this be considered faithful or unfaithful?

$\bullet$ The experimental conclusion that “current LLMs treat intermediate structures as context rather than true mediators of decision making” has already been supported by prior work. Similar findings  have shown inconsistency between generated chains-of-thought and the model’s actual reasoning processes [1–6].

[1] J. Chua et al. Are deepseek r1 and other reasoning models more faithful? arXiv preprint arXiv:2501.08156, 2025.

[2] B. Baker et al. Monitoring reasoning models for misbehavior and the risks of promoting obfuscation. arXiv preprint arXiv:2503.11926, 2025.

[3] Y. Chen et al. Reasoning models don’t always say what they think. arXiv preprint arXiv:2505.05410, 2025.

[4] A. Zhang et al. Reasoning models know when they’re right: Probing hidden states for self-verification. arXiv preprint arXiv:2504.05419, 2025.

[5] M. Turpin et al. Language models don’t always say what they think: Unfaithful explanations in chain-of-thought prompting. NeurIPS, 2023.

[6] I. Arcuschin et al. Chain-of-thought reasoning in the wild is not always faithful. arXiv preprint arXiv:2503.08679, 2025.

**Questions:**

See weaknesses.

---

> ### Author Response · Authors · 2025-11-23
>
> Thank you for the dedicated time! We are glad that you recognize the practical significance of exploring faithfulness of intermediate structures. We would like to address the concerns underlined in your review:
>
> > For autoregressive language models, what exactly does the intermediate reasoning representation $M \neq M^* $ refer to?
>
> We consider semantic, not token-level variation. Semantic inconsistency $M \neq M^* $ is task-dependent, and detailed in Section 3 for each dataset. For example, the RiceChem dataset considers a chemistry grading task, where each example contains a question, a student’s free-form answer and a filled evaluation rubric of True/False items with real-valued weights (concrete examples are given in Appendix A.1, few-shot section of prompt). Final score is defined as the weighted sum of satisfied items. To model a significant discrepancy $M \neq M^* $, we flip the value of one or more of the rubrics, which causally affects the final score. Similarly, in all other datasets interventions on the intermediate structure $M$ are specifically designed to guarantee a change in the expected answer.
> > If semantic-level, how is semantic inconsistency defined and quantified?
> Quantifying semantic inconsistency is indeed a nuanced task. Thus, when we define quantitative faithfulness metrics in Section 2.2, we consider either truly minor changes such as paraphrases (soft variant of HSVT intervention), or specifically designed dataset-specific interventions which guarantee change in answer.
>
> > How is unfaithfulness defined?
>
> As we indicate in the paper title, Section 2.1 and Figure 2, we consider faithfulness to intermediate structures. While in general it it true that arbitrary change $M \neq M^* $ does not necessarily imply a change in the final answer, we specifically design the structural interventions for each task in a way which guarantees a change in the final answer.
>
> > Measuring causality through output change is valid only under restricted settings… Even in tasks with well-defined causal structures, modifying $M$ may not necessarily change the output $Y$.
>
> We agree that an arbitrary change in $M$ does not necessarily imply a change in the output $Y$. However, as discussed above, we address this via task-specific intervention design: our local and global edits do guarantee a change in $Y$. We also agree that for more open-ended tasks causality becomes harder to define, but we still believe there is value in exploring causality for domains with simpler output form, such as text classification (which is re-purposed for grading in RiceChem), context-based fact verification (Averitec), logical inference (EntailmentBank) or query verification (TabFact).
>
> > The paper should provide explicit mathematical definitions and quantitative measures for key concepts. What is LLM unfaithfulness, and how is it measured? What qualifies as a logically significant edit, and how are modifications to $M$ quantified? How is causality defined across tasks, and how can it be measured?
>
> We provide mathematical formalism for our definition of faithfulness to intermediate structures in Section 2.1 (including Figure 2), quantitative measures of unfaithfulness in Section 2.2 (HSVT, LEC, GEC), and elaborate what qualifies as a logically significant edit in Section 3 (since this is inherently task-dependent). Essentially we base our faithfulness metrics on necessary conditions for causal dependency: we know that our interventions must imply $Y != Y^* $, so when the models fail to change their output, this proves unfaithfulness to intermediate structure.
>
> > The authors argue that “If an LLM is faithful to its intermediate structures… then logically significant edits should change its final decision.” This statement is an assumption, not a fact. Modifications to intermediate structures may alter the probability distribution of the model's output, but they do not necessarily change the final decision. For instance, the top-probability token might remain the same even if its confidence decreases sharply. Should this be considered faithful or unfaithful?
>
> Validity of this statement depends on the definition of faithfulness and “logically significant” edits. Our definition of faithfulness is given in Section 2.1, and logically significant edits are discussed in Section 3. In our case, aforementioned statement is a direct implication. For simplicity, we base our definition in terms of model’s predictions rather than probability distributions over predictions. Thus we indeed do not favor models which adjust confidence in the right direction from ones which keep the original answer confidently, and classify both of them as unfaithful.

---

> ### Author Response · Authors · 2025-11-23
>
> > The experimental conclusion that “current LLMs treat intermediate structures as context rather than true mediators of decision making” has already been supported by prior work.
>
> Thank you for bringing up these related works. We will add and discuss them in Section 4.
> We would like to underline following distinctions:
> * Prior studies intervene only on free-form reasoning text. We consider tasks with structured mediators — rubrics, checklists, proofs and verifier queries — which are a priori expected to have greater causal effect than raw text.
> * Second, structured mediators allow for fine-grained, well-defined changes, as opposed to free-form text edits.
> * Finally, we compare settings with gold and predicted intermediate structures, explicitly comparing faithfulness to self-generated vs external context, which has not been done before.
>
> ---
>
> Thanks you again for your comments! We’ll be glad to answer any other questions.

---

> > ### Author Response · Authors · 2025-11-27
> >
> > Dear Reviewer, we have updated our PDF with your concerns and questions, please let us know if you still have any concerns or questions, and whether our comments have affected your overall assessment of our paper.

---

### Meta-Review · Area_Chair_NiZS · 2026-01-05

**Summary:**

The paper looks into whether LLM reasoning can be unfaithful/difficult to “steer”, extending beyond prior work by focusing specifically on structured intermediates (rubrics, checklists, logical graphs), and rigorously testing their causal role. In particular, the authors change intermediate reasoning structures in a way that should change the output , and evaluate LLM’s output under these changes. The research presented shed light on some aspects of LLM interpretability and human-AI collaboration.

The authors need to invest a bit more work into thinking about strong responses to questions coming from Reviewers NWDA and zgVw.  I would encourage the authors to be more rigorous, as their current rebuttal relies heavily on philosophical definitions of collaboration rather than empirical or mathematical justification for why "robustness" to corrupted structures should be penalized as "unfaithfulness".

In general, what's lacking is a deeper self-critique regarding prompting. While the authors added a  "detialed prompt" experiment in the rebuttal, the resulting regression in faithfulness warrants a much deeper analysis of why prompt interventions fail. Future versions would benefit from synthetic cases that bypass the ambiguity of semantic edits, allowing for a cleaner test of the causal mechanism.

**Reviewer Concerns:**

Reviewer NWDA.  Concerns addressed: on clarifying whether the difference between the original and intervened structured was semantic; on novelty compared to CoT faithfulness analysis; on definitions of various concepts. Other concerns were only partially addressed. For example, the review was questioning whether LLM not flipping the response after the changes in the intermediate structure actually indicates unfaithfulness.

 Reviewer dryL. The authors provided a comprehensive rebuttal, running new experiments (e.g., different prompting) and providing statistical analyses as requested by the reviewer. I believe the concerns were addressed.

Reviewer zgVw. Several concerns were addressed (around the definition of expected output in contradictory scenarios, the distinction between different levels of interventions, soft vs global, as well as clarification around some outputs). Others were only partially addressed. For example, the reviewer said that "If the edits are introducing a logical contradiction then... it isn't clear that the model should be expected to answer according to the content of the rubric". They argued that ignoring a corrupted structure to give the correct answer might be a desirable trait (robustness) rather than a flaw. The authors said that what the reviewer calls robustness, they label it as unfaithfulness. The reviewer also noted that the submission's significance "rests heavily on whether the observed behavior is surprising/unexpected".

Reviewer ewGT: pointed out several issues, like missing error bars, wanted clarification on in vs ood data. They also noted that on the EntailmentBank dataset, faithfulness to "gold" structures seemed inversely related to faithfulness to predicted structures. No remaining concerns post-rebuttal.

**Reviewer Scores:**

Reviewer NWDA: would expect this reviewer to lean towards rejection. As I mentioned above, the responses to several concerns were somewhat philosophical.

Reviewer dryL: would expect this reviewer to increase the score.

Reviewer zgVw: while several important concerns were addressed, others remained outstanding. I would expect this to remain a rejection.

Reviewer ewGT: would expect to maintain a high score.

---

### Decision · Program_Chairs · 2026-01-26

Reject